# Growing Neural Gas with Different Topologies for 3D Space Perception

**Yuichiro Toda * , Akimasa Wada, Hikari Miyase , Koki Ozasa, Takayuki Matsuno and Mamoru Minami**

Graduate School of Natural Science and Technology, Okayama University, 3-1-1 Tsushima-Naka, Kita-Ku, Okayama 700-8530, Japan; p3zs7vzv@s.okayama-u.ac.jp (A.W.); pmrj6xn3@s.okayama-u.ac.jp (H.M.); pvta6il3@s.okayama-u.ac.jp (K.O.); matsuno@okayama-u.ac.jp (T.M.); minami-m@cc.okayama-u.ac.jp (M.M.)
* Correspondence: ytoda@okayama-u.ac.jp; Tel.: +81-86-251-8924

**Abstract:** Three-dimensional space perception is one of the most important capabilities for an autonomous mobile robot in order to operate a task in an unknown environment adaptively since the autonomous robot needs to detect the target object and estimate the 3D pose of the target object for performing given tasks efficiently. After the 3D point cloud is measured by an RGB-D camera, the autonomous robot needs to reconstruct a structure from the 3D point cloud with color information according to the given tasks since the point cloud is unstructured data. For reconstructing the unstructured point cloud, growing neural gas (GNG) based methods have been utilized in many research studies since GNG can learn the data distribution of the point cloud appropriately. However, the conventional GNG based methods have unsolved problems about the scalability and multi-viewpoint clustering. In this paper, therefore, we propose growing neural gas with different topologies (GNG-DT) as a new topological structure learning method for solving the problems. GNG-DT has multiple topologies of each property, while the conventional GNG method has a single topology of the input vector. In addition, the distance measurement in the winner node selection uses only the position information for preserving the environmental space of the point cloud. Next, we show several experimental results of the proposed method using simulation and RGB-D datasets measured by Kinect. In these experiments, we verified that our proposed method almost outperforms the other methods from the viewpoint of the quantization and clustering errors. Finally, we summarize our proposed method and discuss the future direction on this research.

**Keywords:** 3D space perception; growing neural gas; topological structure learning method

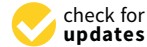



## 1. Introduction

Three-dimensional space perception is one of the most important capabilities for an autonomous mobile robot in order to operate a task in an unknown environment adaptively since the autonomous robot needs to detect the target object and estimate the 3D pose of the target object for performing given tasks efficiently [1–7]. In the research field of 3D space perception, many research studies use an RGB-D camera, such as Microsoft Kinect [8] and Intel Realsense [9], that can measure the color and depth of an image simultaneously in real time. After the 3D point cloud is measured, the autonomous robot needs to reconstruct a structure from the 3D point cloud with color information according to the given tasks since the point cloud is unstructured data.

For realizing the 3D space perception, many kinds of 3D image processing based methods have been proposed. These methods were expanded to the 3D point cloud by utilizing camera image processing technologies, such as filtering methods [10], feature extractions [11,12], and sample consensus methods [13]. However, these methods depend on feature or model descriptors, which have a problem with adaptability to environmental changes, such as geometry, texture and light conditions. To improve the adaptability, recently, the research studies of the information extraction from the point cloud were

developed by a deep neural network (DNN) [14]. In particular, many research works in autonomous robot and automatic driving have proposed a DNN-based semantic segmentation method (e.g., PointNet [15], PointCNN [16] and VoxNet [17]) that can label and give the point cloud meaning. The DNN-based methods give highly accurate segmentation results by using big data with the teaching signal. However, one of the problems of the supervised learning method is application to unknown data. If the learned network is applied to the unknown environment or label not including the output layer, the network fails to segment the point cloud. In the environment where the autonomous robots are expected to perform the task, the autonomous robot needs to recognize the unknown target object and environment. Therefore, the point cloud processing method without prior knowledge is required for realizing the robot that can perform the given tasks in the unknown environment.

Self-organization map (SOM) [18] based methods are one of the main streams based on unsupervised learning for the point cloud processing [19]. The other methods that are well known in this field are neural gas (NG) [20], growing cell structure (GCS) [21], and growing neural gas [22]. Basically, these methods are called unsupervised learning when applied to an unknown data distribution without teaching signals and using the competitive learning method based on the winner-take-all approach. The SOM can project the input vectors in a high dimensional space to a topological structure in a low dimensional space, according to the data distribution. The SOM is easy to apply to many problems by determining the number of nodes and topologies, while it is difficult to design these things to the unknown data distribution. NG does not need to determine the number of nodes since the NG has node addition and deletion in the learning algorithm. However, the topological structure is updated according to the order of the data input. On the other hand, GCS and GNG can dynamically change the topological structure based on the adjacency relation held in the referred node. GCS cannot delete the nodes and edges, while GNG can dynamically delete the nodes and edges by using the concept of the edge's age. Figure 1 shows an example result of each method applied to a 2D simulation point cloud. There are redundant nodes and edges in SOM and GCS , while GNG can preserve the 2D point cloud space appropriately. In addition, GNG can cluster the unknown ring data (Rings A, B and C) by utilizing the topological structure since there is no adjacent relations between each ring (Figure 1e). Moreover, GNG can perform noise reduction [23,24], 3D reconstruction [25–28] and feature extraction using the topological structure [29,30]. From these reasons, GNG is expected to utilize the unified perceptual system for the point cloud processing [31–39]. D. Viejo et al. applied GNG-based 3D feature extraction and the matching method to 3D object recognition [40]. This method just utilizes the node set for extracting features such as SHOT and spin image, which do not utilize the topological structure generated by GNG. Refs. [41–44] reduced the calculation cost and improved the adaptability of the time-series data for realizing gesture recognition and 3D object-tracking methods in real time. Ref. [41] applied a uniform grid structure to GNG for reducing the calculation cost of the winner node selection and realized the real-time 3D object-tracking method. Refs. [43,44] proposed the criteria for node addition and deletion. These methods are based on the probability density distribution of the data and nodes and verified the real-time adaptability for the point cloud data.

In this way, many kinds of GNG-based methods have been proposed from the viewpoint of various research directions. However, the GNG-based perceptual system has unsolved problems. One of the problems is that GNG cannot preserve the 3D position space if the input vector is composed of not only the 3D point cloud (position information), but also additional information, such as RGB-D data (3D position with color information). Figure 2 represents a learning example using GNG from a 3D point cloud ($x$, $y$, $z$) with color information ($R$, $G$, $B$). The generated topological structure shown in Figure 2b cannot preserve the 3D position space of the point cloud since the topological structure is generated from both the position and color information, and the scale of the color information is a dominant scale compared with the scale of the position information in this dataset. In this

way, GNG-based point cloud processing methods have the scalability problem between each feature vector if the input vector is composed of multiple properties.

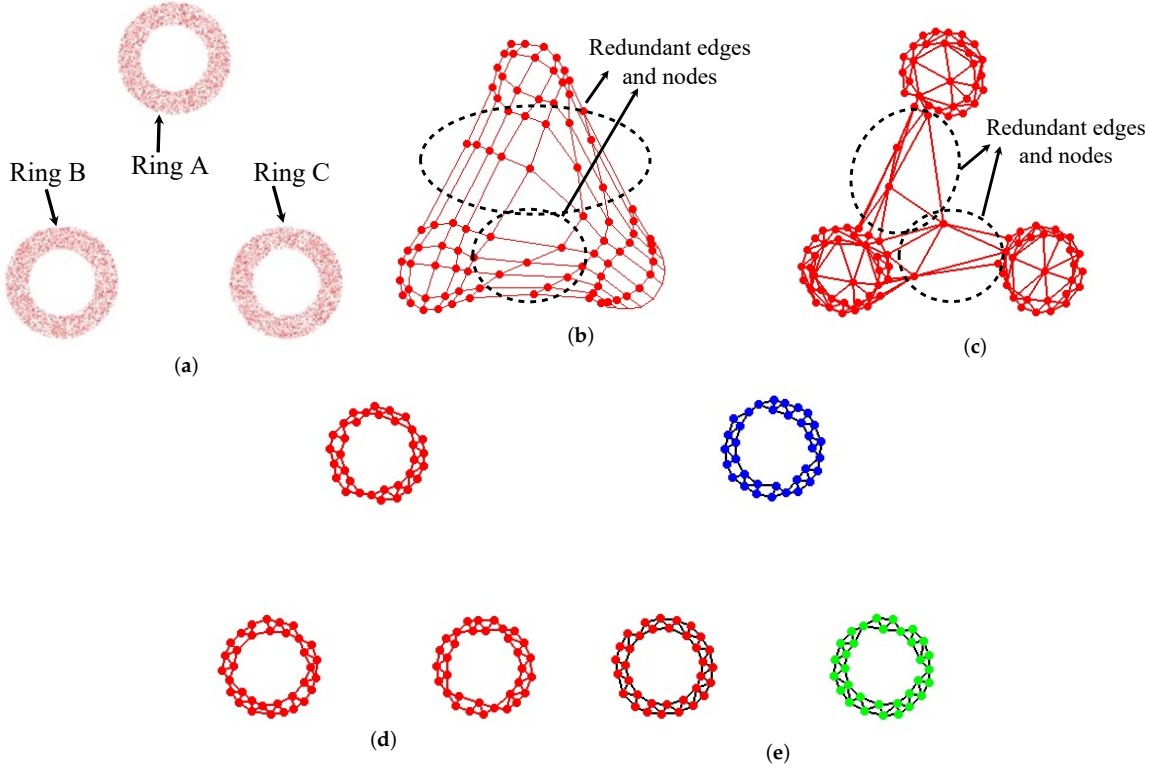

**Figure 1.** Examples of topological learning methods. In (**e**), each color indicates each cluster. (**a**) Input data, (**b**) SOM, (**c**) GCS, (**d**) GNG, (**e**) clustering.



**Figure 2.** An example of GNG that applied to simulation data. Circles and lines respectively indicate node and edge of GNG in (**b**,**c**). (**a**) Image data, (**b**) result of GNG, (**c**) result of GNG with the weighted vector.

For solving this problem, there exist two main approaches. One is to utilize a background subtraction algorithm as preprocessing for extracting the target data and generating the topological structure of only the target object [45,46]. The other is to apply a weighted distance measurement in the winner node selection according to the importance of each property [47,48]. In the former approach, Angelopoulou, et al. proposed a background detection method based on mixture Gaussian distribution and CIE lab color space for generating topological structure of the human face and arm [45]. However, this kind of approach cannot preserve the appropriate position space if the background detection fails because of the sensor noises. In addition, the background detection needs detailed prior knowledge about the target object. Therefore, it is difficult to apply the background detection method to an unknown dataset. In the latter approach, Tunnermann, et al. proposed the GNG attention method (GNGA) for extracting saliency from a 2D camera image [47]. GNGA uses the weighted distance measurement used in the selection of the 1st and 2nd winner nodes for generating the topological structure of the position space with color

information. In addition, the authors also proposed the weight-vector-based GNG method, called modified GNG with utility (GNG-U II) [48]. These methods can preserve the 2D/3D position spaces with color information by designing the appropriate weight.

Figure 2c shows an example result of GNG with the weighted distance measurement. By using the weighted distance measurement, the topological structure represents the original 3D environmental space of the point cloud and learns the color information simultaneously. In particular, the algorithm of GNGA includes the normalization of each feature vector in the input vector for solving the scalability problem in the 2D image. However, it is difficult to normalize the input vector of the 3D point cloud with the other properties if the point cloud is composed of the increasing environmental map. In addition, the number of clusters generated from the topological structure is only one since it is difficult to cut the edges between each objects from the floor surface (in the green circles of Figure 2c) by designing the appropriate weights from the unknown data distribution composed of the multiple properties' vector. Therefore, the weighted distance-based methods need a cutting algorithm of the edges for clustering the 2D/3D point cloud data according to the property after generating the topological structure. In this way, the learning method of the 2D/3D point cloud with additional information that can generate the topological structure composed of the multiple properties and preserve the position space simultaneously is not realized.

In this paper, we propose growing neural gas with different topologies (GNG-DT) as a new topological structure learning method for addressing all of these issues, which were not incorporated in the previous work. GNG-DT has multiple topologies of each property, while the conventional GNG method has a single topology of the input vector. In addition, the distance measurement in the winner node selection uses only the position information for preserving the environmental space of the point cloud. The main contributions of this paper are listed as follows:

1. GNG-DT can preserve 2D/3D position spaces with additional feature information from the point cloud by using the distance measurement of only the position information.
2. GNG-DT can give multiple clustering results by utilizing multiple topologies within the framework of online learning.
3. GNG-DT is a robust learning algorithm for scale variance of the input vector composed of point cloud with additional properties.

This paper is organized as follows. Section 2 proposes growing neural gas with different topologies for the 3D space perception. Section 3 shows several experimental results by using 2D and 3D simulation data and RGB-D data. Finally, we summarize this paper and discuss the future direction to realize the 3D space perception.

## 2. Growing Neural Gas with Different Topologies
### 2.1. Overview of Algorithm

In this section, we explain the overview of our proposed method called "growing neural gas with different topologies (GNG-DT)". Figure 3 shows an overall image of the GNG-DT-based point cloud processing method. GNG-DT uses almost the same distance measurement as GNG-U II, and GNG-DT learns the multiple topologies for clustering the point cloud from the different viewpoints of the multiple properties. First, we define the variables used in GNG-DT.

The set $S^{in}$ of the input vector's properties is $S^{in} = \{$Position $(pos)$, Color $(col)$, ...$\}$; the set $S^{ref}$ of the node's properties is $S^{ref} = \{$Position $(pos)$, Color $(col)$, Normal vector $(nor)$, ...$\}$; and the input vector $\mathbf{v}$ and the $i$th node (reference vector) $\mathbf{h}$ are defined as $\mathbf{v} = (\mathbf{v}^{pos}, \mathbf{v}^{col}, ...)$, $\mathbf{h}_i = (\mathbf{h}_i^{pos}, \mathbf{h}_i^{col}, \mathbf{h}_i^{nor}, ...)$, respectively. Next, a distance measurement of the $o$th property between the input vector and the $i$th node is defined as

$$d_i^o = \|\mathbf{v}^o - \mathbf{h}_i^o\| \tag{1}$$

In addition, the edge set of the $o$th property is defined as $C^o = \{c^o_{1,2}, \ldots, c^o_{i,j}, \ldots\}$ ($\forall o \in S^{ref}$) for generating the multiple topologies in GNG-DT. The detailed algorithm is explained as follows.

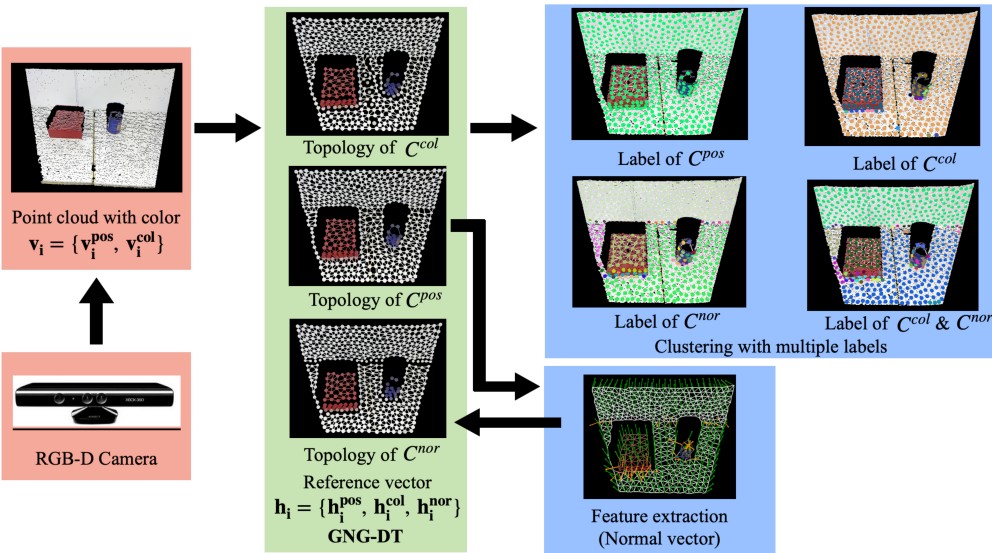

**Figure 3.** Overview of the major steps in GNG-DT. After measuring 3D point cloud with color from RGB-D camera, the multiple topologies are generated by using competitive learning method. By utilizing and combining the multiple topologies, GNG-DT can provide various clustering results from the viewpoint of each property.

**Step 0.** Generate two nodes at random positions, $\mathbf{h}_1$ and $\mathbf{h}_2$ in $R^n$, where $n$ is the dimension of the reference vector. Initialize the connection set $c^o_{1,2} = 1$ ($\forall o \in S^{ref}$), and the age of edge $g_{1,2}$ is set to 0.

**Step 1.** Generate at random an input data $\mathbf{v}$.

**Step 2.** Select the 1st winner node $s_1$ and the 2nd winner node $s_2$ from the set of nodes by

$$s_1 = \arg\min_{i \in A} d^{pos}_i \tag{2}$$

$$s_2 = \arg\min_{i \in A \setminus s1} d^{pos}_i$$

where $A$ indicates the set of the node numbers.

**Step 3.** Add the squared distance between the input data and the 1st winner to an accumulated error variable $E_{s_1}$:

$$E_{s_1} \leftarrow E_{s_1} + (d^{pos}_{s_1})^2 \tag{3}$$

**Step 4.** Set the age of the connection between $s_1$ and $s_2$ at 0 ($g_{s_1,s_2} = 0$). If a connection of the position information between $s_1$ and $s_2$ does not yet exist, create the connection ($c^{pos}_{s_1,s_2} = 1$). In addition, the edge of the other property $o(\in S^{ref} \setminus pos)$ is calculated as the following equation:

$$\begin{cases} c^o_{s_1,s_2} = 1 & if \; \|\mathbf{h}^o_{s_1} - \mathbf{h}^o_{s_2}\| < \tau^o \\ c^o_{s_1,s_2} = 0 & otherwise \end{cases} \tag{4}$$

where $\tau^o$ is the predefined threshold value of the $o$th property.

**Step 5.** Update the nodes of the winner and its direct topological neighbors by the learning rates $\eta_1$ and $\eta_2$ ($\eta_1 > \eta_2$).

$$\begin{aligned}
\mathbf{h}_{s_1} &\leftarrow \mathbf{h}_{s_1} + \eta_1(\mathbf{v} - \mathbf{h}_{s_1}) \\
\mathbf{h}_j^o &\leftarrow \mathbf{h}_j^o + \eta_2(\mathbf{v} - \mathbf{h}_j^o) \quad if \;\; c_{s_1,j}^o = 1
\end{aligned} \tag{5}$$

**Step 6.** Increment the age of all edges emanating from $s_1$.

$$g_{s_1,j} \leftarrow g_{s_1,j} + 1 \quad if \;\; c_{s_1,j}^{pos} = 1 \tag{6}$$

**Step 7.** Delete edges of all properties with an age larger than $g_{max}$. If this results in nodes having no more connecting edges ($c_{s_1,s_2}^{pos} = 0$) of the position information, remove those nodes as well.

**Step 8.** If the number of input data generated so far is an integer multiple of a parameter $\lambda$, insert a new node as follows.

i. Select the node $u$ with the maximal accumulated error.

$$u = \arg \max_{i \in A} E_i \tag{7}$$

ii. Select the node $f$ with the maximal accumulated error among the neighbors of $u$.

iii. Add a new node $r$ to the network and interpolate its node form $u$ and $f$.

$$\mathbf{h}_r = 0.5(\mathbf{h}_u + \mathbf{h}_f) \tag{8}$$

iv. Delete the original edges of all properties between $u$ and $f$. Next, insert edges of the position property connecting the new node $r$ with nodes $u$ and $f$ ($c_{u,r}^{pos} = 1$, $c_{r,f}^{pos} = 1$). The edges of the $o$th property ($\in S^{ref} \backslash pos$) are calculated as the following equation:

$$\begin{cases} c_{i,j}^o = 1 & if \; \|\mathbf{h}_i^o - \mathbf{h}_j^o\| < \tau^o \\ c_{i,j}^o = 0 & otherwise \end{cases} \tag{9}$$

v. Decrease the error variables of $u$ and $f$ by a temporal discounting rate $\alpha(0 \leq \alpha \leq 1)$.

$$\begin{aligned}
E_u &\leftarrow E_u - \alpha E_u \\
E_f &\leftarrow E_f - \alpha E_f
\end{aligned} \tag{10}$$

vi. Interpolate the error variable of $r$ from $u$ and $f$.

$$E_r = 0.5(E_u + E_f) \tag{11}$$

**Step 9.** Decrease the error variables of all nodes by a temporal discounting rate $\beta$ ($0 \leq \beta \leq 1$).

$$E_i \leftarrow E_i - \beta E_i \quad (\forall i \in A) \tag{12}$$

**Step10.** Continue with step 1 if a stopping criterion (e.g., the number of nodes or some performance measure) is not yet fulfilled.

　　Figure 4 shows the total algorithm of GNG-DT, and bold squares indicate the difference part of the conventional method of the GNG algorithm. In the following sections, we explain the difference points of the conventional algorithm.

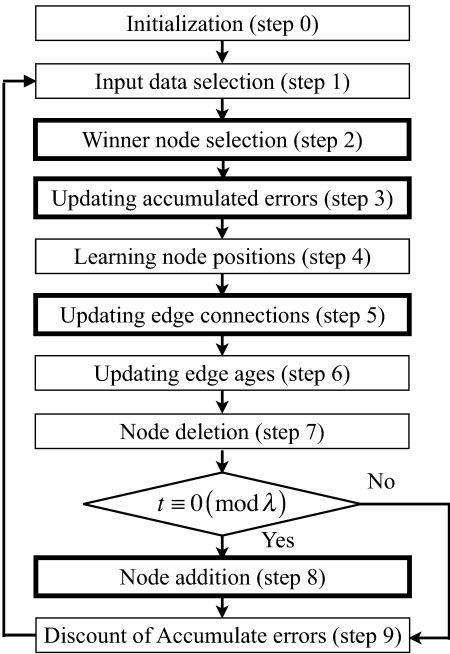

**Figure 4.** Flowchart of GNG-DT, where t indicates the number of data input. The boxes drawn by bold line indicate the modified step from the conventional GNG.

*2.2. Distance Measurement*

In the conventional methods, such as GNGA and GNG-U II, the weighted distance measurement is used in the winner node selection and accumulated error update as the following equation:

$$s_1 = \arg\min_{i \in A} \sum_{o \in S} w^o d_i^o \tag{13}$$

$$s_2 = \arg\min_{i \in A \setminus s_1} \sum_{o \in S} w^o d_i^o$$

$$E_{s_1} \leftarrow E_{s_1} + \sum_{o \in S} w^o (d_{s_1}^o)^2 \tag{14}$$

where $w^o$ indicates the weight of the $o$th property. In the weighted distance methods, it is difficult to design the weight for preserving the 3D environmental space appropriately if the input vector is composed of the point cloud and the other feature vectors because the scalability of each feature is different. On the other hand, the distance measurement of GNG-DT uses only the position information of the point cloud shown in Equations (2) and (3) that mean $w^{pos} = 1$ and $w^o = 0$ ($o \in S^{in} \setminus pos$) in Equations (13) and (14) for learning the accurate position space of the point cloud.

*2.3. Clustering from Multiple Topologies*

The conventional methods generate only one topology, and the edge is added to the topology if there is no edge between the 1st winner node $s_1$ and the 2nd winner node $s_2$. Therefore, the conventional methods need to use the cutting edge algorithm for clustering the point cloud data after generating the topological structure. On the other hand, GNG-DT generates the multiple topologies of each property including in the property set of the reference vector ($S^{ref}$). Therefore, GNG-DT needs to add the edges with each feature, except the position information ($c_{s1,s2}^{pos}$) for generating the multiple topologies. The edges of the other properties are added by calculating the similarity distance between the nodes of each feature and using the predefined threshold value. Figure 5a–c shows the concept image of

the multiple topologies in the point cloud with color information. In this case, GNG-DT can generate three topologies composed of position, color and normal vector.

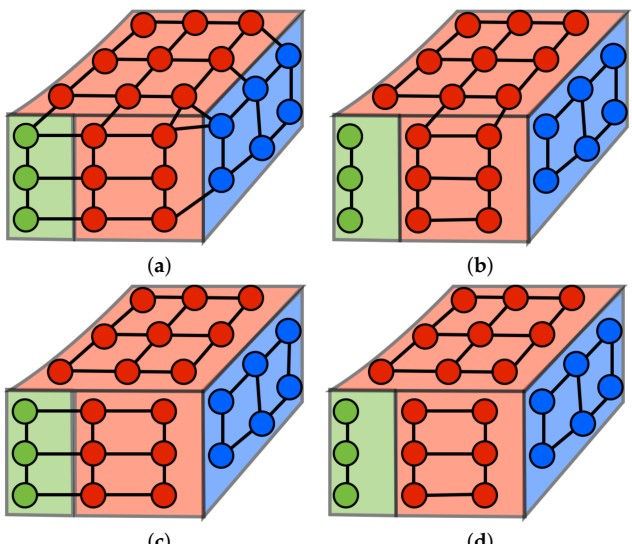

**Figure 5.** Examples of multiple topologies of GNG-DT. By combining each topological structure, GNG-DT can provide additional clustering results, such as (**d**). (**a**) Position information, (**b**) color information, (**c**) normal vector information, (**d**) color and normal vector information.

In the clustering of GNG-DT, the topological structure of each property is utilized by searching the connectivity of each cluster. In addition, GNG-DT can provide additional clustering results by combining the multiple topologies. Figure 5d shows an example of the combination using the topologies of the color and normal vector information ($c_{i,j}^{col} \cdot c_{i,j}^{nor} = 1$). In this way, the multiple topologies enable to provide the multiple clustering results according to the number of property sets of the reference vector and the combination of each property.

### 2.4. Learning Rule

Next, we explain the learning rule of GNG-DT. The conventional learning rule is calculated as the following equation:

$$
\begin{aligned}
\mathbf{h}_{s_1} &\leftarrow \mathbf{h}_{s_1} + \eta_1(\mathbf{v} - \mathbf{h}_{s_1}) \\
\mathbf{h}_j &\leftarrow \mathbf{h}_j + \eta_2(\mathbf{v} - \mathbf{h}_j) \;\; if \;\; c_{s_1,j} = 1
\end{aligned}
\tag{15}
$$

where $c_{s_1,j}$ indicates the edge of the conventional GNG method, and $\eta_1$ and $\eta_2$ indicate learning rates. GNG-DT also uses this learning rule for the 1st winner node $s_1$. On the other hand, the topological neighborhood of the 1st winner node is updated if the edge of each property is connected (Equation (5)). In this way, the winner node selection and accumulated error calculation use the distance measurement of only the position information, while the node update uses all of the properties for learning the geometric space and the other feature vectors from the point cloud.

### 2.5. Feature Extraction from the Topological Structure

While learning the topological structure, the normal vectors are extracted as one of the properties of the reference vector ($\mathbf{h}_i^{nor}$) from the reference vector of the position information ($\mathbf{h}_i^{pos}$). Figure 6 shows the concept image of the local surface. At first, a local surface (depicted in green circles in Figure 6) of the *i*th node is composed of the nearest

nodes ($c_{i,j}^{pos} = 1$). Next, the weighted center of gravity of the local surface is calculated, and then covariance matrix $\bar{\mathbf{h}}_i^{pos}$ is calculated as follows.

$$
\begin{aligned}
Cov_i &= \sum_{j=1}^{k} (\mathbf{h}_j^{pos} - \bar{\mathbf{h}}_i^{pos})^T (\mathbf{h}_j^{pos} - \bar{\mathbf{h}}_i^{pos}) \\
\bar{\mathbf{h}}_i^{pos} &= \frac{\mathbf{h}_i^{pos} + \sum_{j=1}^{k} \omega_{i,j} \cdot \mathbf{h}_j^{pos}}{1 + \sum_{j=1}^{k} \omega_{i,j}} \\
\omega_{i,j} &= exp(-\frac{||\mathbf{h}_i^{pos} - \mathbf{h}_j^{pos}||^2}{\mu^2})
\end{aligned}
\tag{16}
$$

where $\mu$ indicates the coefficient. After calculating the covariance matrix, the eigenvectors and values of the matrix are calculated for estimating the normal vector. Then, the normal vector $\mathbf{h}_i^{nor}$ is assigned to the eigenvector with the minimum eigenvalue [11]. In this way, the property of the reference vector cloud is calculated by utilizing the topological structure of GNG-DT.

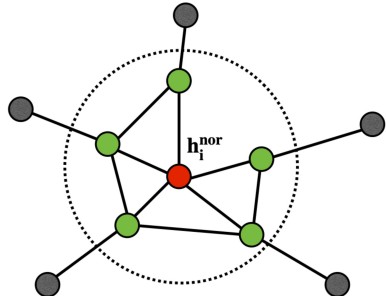

**Figure 6.** Concept image of the local surface for extracting normal vector. The surface elements represent green nodes in the dot circle.

## 3. Experimental Results

### 3.1. Simulation Data

3.1.1. Experimental Setup

In this section, we conduct simulation experiments using 2D and 3D point cloud data for verifying the effectiveness of our proposed method. Figure 7 shows the dataset in this experiment. In Figure 7a, the total number of the 2D point cloud composed of the red and green rectangles is 10,000 and the dimensions are (width 300) $\times$ (height 300). The centers $(x, y)$ of the two red and green rectangles C1, C2, C3, and C4 are (75, 75), (225, 75), (75, 225), and (225, 225), respectively. In Figure 7b, the total number of the 3D point cloud is 112,500, and the point cloud is composed of a half of a red/brown quadratic prism, a half of a red cylinder, and a brown surface.

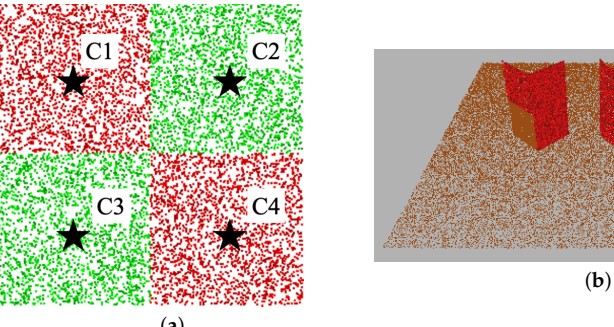

(a)

(b)

**Figure 7.** Experimental dataset. (**a**) 2D point cloud (Data1), (**b**) 3D point cloud (Data2).

The distance measurement of GNG-DT in the winner node selection uses only the position information. Therefore, the scale variance of the position information affects the learning result. In this experiment, the scale factor ($scl$ = 10.0, 1.0, 0.1, 0.01) multiplied $\mathbf{v}^{pos}$ of each dataset for verifying the scale variance in GNG-DT. The input vector is composed of $\mathbf{v} = (scl \cdot \mathbf{v}^{pos}, \mathbf{v}^{col})$. In this experiment, we compared our proposed method with the conventional GNG and GNG with weighted distance measurement (GNG-W) used in GNGA and GNG-U II. GNG-W does not use the normalization of the point cloud for verifying the robustness of the scale variance. The weights of GNG-W are set to $w^{pos} = 0.8$ and $w^{col} = 0.2$, the parameters used in [47]. In addition, Table 1 shows the parameters used in this experiment. These parameters were also used in [47].

**Table 1.** Setting parameters.

| | |
|---|---|
| $\lambda$ | 500 |
| $\eta_1$ | 0.025 |
| $\eta_2$ | 0.0003 |
| $g_{max}$ | 88 |
| $\alpha$ | 0.5 |
| $\beta$ | 0.0005 |
| $\tau^{col}$ | 10.0 |
| $\tau^{nor}$ | 0.02 |

3.1.2. Quantitative Evaluation

In this experiment, we define the following evaluation for verifying the effectiveness of the proposed method.

(1)  Quantization error

The quantization error of each property is defined as follows,

$$r^o = \sqrt{\frac{1}{N} \sum_{i=1}^{N} \|\mathbf{v}_i^o - \mathbf{h}_{s_1}^o\|^2} \tag{17}$$

$$s_1 = \arg\min_{j \in A} \|(\mathbf{v}_i^{pos} - \mathbf{h}_j^{pos})\|$$

where $s_1$ indicates the 1st winner node of the position information. By using Equation (17), we can evaluate the accuracy of the node position and color information included in reference nodes.

(2)  Error of the position information between the center of cluster and true value

Data1 is composed of four rectangles (two red and two green rectangles). The error value between the center of each cluster and true value is defined as follows:

$$e_i = \|\mathbf{m}_i^{pos} - \mathbf{c}_i\| \tag{18}$$

where $\mathbf{m}_i^{pos}$ indicates the center of gravity of the $i$th rectangle's cluster, and $\mathbf{c}_i$ indicates the true position of the $i$th rectangle. Equation (18) evaluates the accuracy of the clustering result in each method. In this experiment, the number of trials in each method is set to 50 since the GNG-based methods use random sampling, and we calculate the average and variance value of each evaluation.

*3.2. RGB-D Data*

3.2.1. Experimental Setup

In this section, we conduct the experiment using 3D point cloud data measured by an RGB-D camera for verifying the effectiveness of the real sensing data. The first column of Figure 12 shows the six datasets used in this experiment. These datasets are measured by

Azure Kinect DK and composed of target objects on the floor or desk. In this experiment, we also compared our proposed method with the conventional GNG and GNG-W, and the quantization error is used as the evaluation metric for preserving the 3D position space of the 3D point cloud. The parameters of each method use the same as the above simulation experiment, and the number of trials is 50.

### 3.2.2. Experimental Result

Tables 2 and 3 show the results of the quantization error, and Figures 8 and 9 show examples of the learning result. In the quantization error of position information, GNG-DT outperformed the other methods for all of the scales, except $scl = 10.0$ in Data2, and GNG-DT preserved the 2D and 3D space structures of each point cloud more accurately. In addition, GNG-DT generated almost the same topological structures of the position ($C^{pos}$) and color ($C^{col}$) properties in the different scales from Figures 8 and 9. The topological structure of color information ($C^{col}$) in Figure 8 was separated by the boundary of the different color, and GNG-DT clustered each rectangle in all scales. On the other hand, GNG and GNG-W generated different topological structures in the different scales. In particular, GNG and GNG-W could not preserve the space structure of the point cloud in the cases of $scl = 0.1$ and $0.01$. This result indicates that the influence of the color information is dominant over the position information. Figure 10 shows that the position information's quantization error multiplied the inverse of the scale factor $scl$. The results of GNG and GNG-W were affected by the scale factor, while GNG-DT did not depend on the scale factor since the error values are almost the same in Figure 10. These results also show that GNG-DT can preserve the space structure of the point cloud without affecting the input vector of the other properties by using only the position information in the distance measurement.

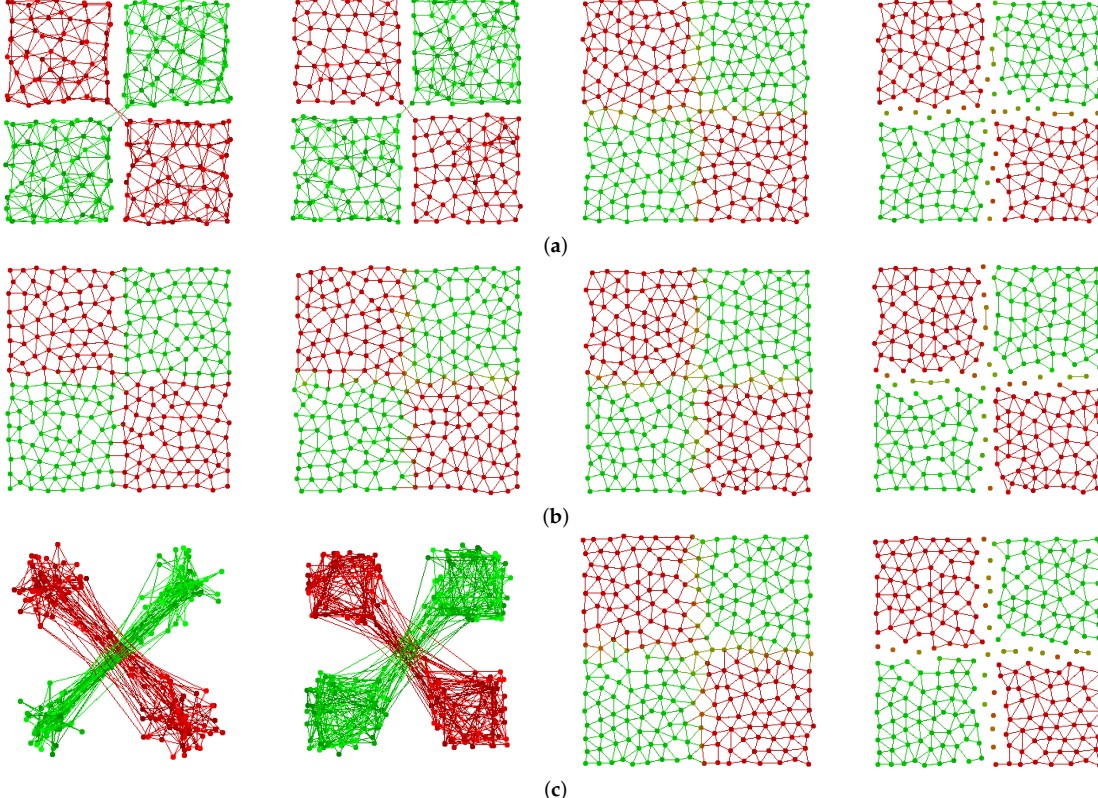

(a)

(b)

(c)

**Figure 8.** Learning examples in simulation experiment using 2D point cloud (Data1). (**a**) $scl = 1.0$ (**Left**: GNG, **Center left**: GNG-W, **Center right**: GNG-DT($C^{pos}$), **Right**: GNG-DT($C^{col}$)), (**b**) $scl = 10.0$ (**Left**: GNG, **Center left**: GNG-W, **Center right**: GNG-DT($C^{pos}$), **Right**: GNG-DT($C^{col}$)), (**c**) $scl = 0.01$ (**Left**: GNG, **Center left**: GNG-W, **Center right**: GNG-DT($C^{pos}$), **Right**: GNG-DT($C^{col}$)).

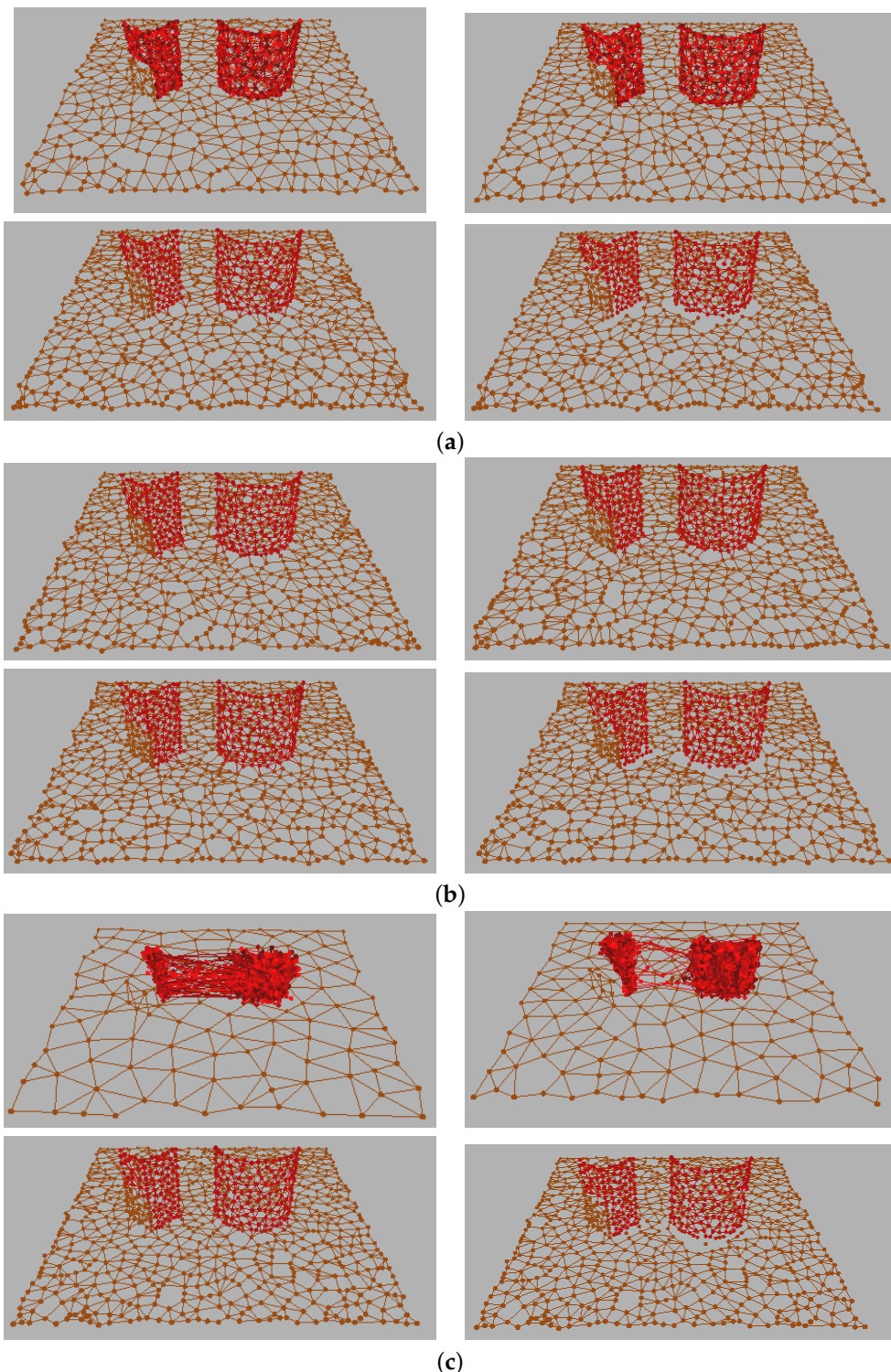

**Figure 9.** Learning examples in simulation experiment using 3D point cloud (Data2). (**a**) $scl = 1.0$ (**Upper left**: GNG, **Upper right**: GNG-W, **Lower left**: GNG-DT($C^{pos}$), **Lower right**: GNG-DT($C^{col}$)), (**b**) $scl = 10.0$ (**Upper left**: GNG, **Upper right**: GNG-W, **Lower left**: GNG-DT($C^{pos}$), **Lower right**: GNG-DT($C^{col}$)), (**c**) $scl = 0.01$ (**Upper left**: GNG, **Upper right**: GNG-W, **Lower left**: GNG-DT($C^{pos}$), **Lower right**: GNG-DT($C^{col}$)).

**Table 2.** Experimental results of quantization error of position in simulation experiment, where GNG, GNG-W and GNG-DT indicate growing neural gas, GNG with weighted distance measurement, and GNG with different topologies, respectively. *scl* indicates scale factor.

| Data1 | GNG | GNG-W | GNG-DT |
|---|---|---|---|
| *scl* = 1.0 | 8.687 $\pm$ 0.062 | 7.794 $\pm$ 0.037 | **7.341 $\pm$ 0.022** |
| *scl* = 10.0 | 73.948 $\pm$ 0.191 | 73.581 $\pm$ 0.217 | **73.434 $\pm$ 0.199** |
| *scl* = 0.1 | 1.311 $\pm$ 0.020 | 1.144 $\pm$ 0.015 | **0.735 $\pm$ 0.002** |
| *scl* = 0.01 | 0.248 $\pm$ 0.009 | 0.190 $\pm$ 0.003 | **0.073 $\pm$ 0.000** |
| Data2 | GNG | GNG-W | GNG-DT |
| *scl* = 1.0 | 11.825 $\pm$ 0.047 | 11.133 $\pm$ 0.038 | **10.341 $\pm$ 0.021** |
| *scl* = 10.0 | **103.064 $\pm$ 0.311** | 103.224 $\pm$ 0.293 | 103.418$\pm$0.322 |
| *scl* = 0.1 | 1.644 $\pm$ 0.010 | 1.456 $\pm$ 0.008 | **1.0349 $\pm$ 0.002** |
| *scl* = 0.01 | 0.279 $\pm$ 0.003 | 0.234 $\pm$ 0.002 | **0.103 $\pm$ 0.000** |

**Table 3.** Experimental results of quantization error of color in simulation experiment, where GNG, GNG-W and GNG-DT indicate growing neural gas, GNG with weighted distance measurement, and GNG with different topologies, respectively. *scl* indicates scale factor.

| Data1 | GNG | GNG-W | GNG-DT |
|---|---|---|---|
| *scl* = 1.0 | 25.152 $\pm$ 0.499 | **22.583 $\pm$ 0.567** | 26.178 $\pm$ 0.425 |
| *scl* = 10.0 | **21.094 $\pm$ 0.554** | 23.081 $\pm$ 0.623 | 26.168 $\pm$ 0.509 |
| *scl* = 0.1 | 28.619 $\pm$ 0.682 | 27.623 $\pm$ 0.650 | **26.072 $\pm$ 0.534** |
| *scl* = 0.01 | 33.693 $\pm$ 1.891 | 30.425 $\pm$ 1.005 | **26.094 $\pm$ 0.475** |
| Data2 | GNG | GNG-W | GNG-DT |
| *scl* = 1.0 | 46.740 $\pm$ 0.290 | 45.396 $\pm$ 0.233 | **33.925 $\pm$ 0.034** |
| *scl* = 10.0 | 34.099 $\pm$ 0.055 | 33.933 $\pm$ 0.028 | **33.919 $\pm$ 0.032** |
| *scl* = 0.1 | 48.084 $\pm$ 0.496 | 47.841 $\pm$ 0.419 | **33.915 $\pm$ 0.031** |
| *scl* = 0.01 | 49.462 $\pm$ 1.301 | 48.056 $\pm$ 0.766 | **33.915 $\pm$ 0.034** |

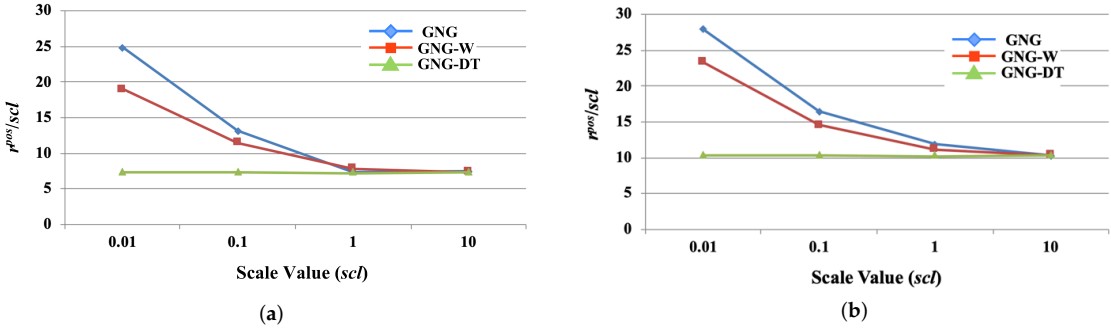

**Figure 10.** Experimental results of quantization errors ($r^{pos}$) divided by scale vales (*scl*). (**a**) Data1, (**b**) Data2.

In the results of the quantization error of color information (Table 3), the error values of GNG and GNG-W were lower than GNG-DT in the case of *scl* = 1.0 and 10.0, while GNG-DT outperformed the other methods in the other results. When the scale factor *scl* is 1.0, the scales of the position and color information are almost the same scale; GNG-W outperformed the other methods. On the other hand, GNG, which can be considered GNG-W whose weights are $w^{pos} = 1.0$, $w^{col} = 1.0$, outperformed the other methods in the case of *scl* = 10.0. In this way, the weighted distance approaches need to design the suitable weight according to the scale of each property for learning the point cloud effectively. On the other hand, GNG-DT, which can be considered to be GNG-W, whose weights are $w^{pos} = 1.0$, $w^{col} = 0.0$, does not depend on the scale factor, and the results are almost the same value.

In addition, the color distribution of the nodes is the same as the point cloud, except the boundary nodes from Figure 8. From these results, GNG-DT can robustly generate the topological structure from the point cloud with color information in the scale differences.

In the result of clustering, the GNG and GNG-W could not cluster four rectangles from the 2D simulation data, while only GNG-DT could cluster four rectangles in all of the scales. Specifically, as the scale factor is 10.0, GNG and GNG-W could not preserve the space structure of the point cloud. This shows that the edges between the same color were not deleted since the color information is a predominant factor in the input vector. On the other hand, the topological structure of GNG-DT ($C^{col}$) could cluster the four rectangles in all of the scale factors. Table 4 shows the experimental result of the error $e_i$ between the center of cluster and true value in each rectangle in the topological structure $C^{col}$. The results of error had constant values in all of the scales. In Figure 8, the topological structures of color information had isolated nodes in the vicinity of the boundary. As a result of the isolated nodes, the centers of clusters shifted from the true value of each center.

**Table 4.** Experimental results of clustering error of GNG-DT, where GNG-DT indicates growing neural gas with different topologies. *scl* indicates scale factor.

| Cluster No. | Cluster Error ($e_i$) | Cluster No. | Cluster Error ($e_i$) |
|---|---|---|---|
| C1 (*scl* = 1.0) | $5.691 \pm 1.520$ | C2 (*scl* = 1.0) | $5.691 \pm 1.809$ |
| C1 (*scl* = 10.0) | $57.654 \pm 15.035$ | C2 (*scl* = 10.0) | $57.562 \pm 14.653$ |
| C1 (*scl* = 0.1) | $0.541 \pm 0.152$ | C2 (*scl* = 10.0) | $57.562 \pm 14.653$ |
| C1 (*scl* = 0.01) | $0.058 \pm 0.019$ | C2 (*scl* = 0.01) | $0.053 \pm 0.014$ |
| C3 (*scl* = 1.0) | $5.518 \pm 1.759$ | C4 (*scl* = 1.0) | $5.830 \pm 1.700$ |
| C3 (*scl* = 10.0) | $56.296 \pm 19.622$ | C4 (*scl* = 10.0) | $57.493 \pm 16.728$ |
| C3 (*scl* = 0.1) | $0.601 \pm 0.191$ | C4 (*scl* = 0.1) | $0.564 \pm 0.165$ |
| C3 (*scl* = 0.01) | $0.055 \pm 0.016$ | C4 (*scl* = 0.01) | $0.056 \pm 0.016$ |

Next, Figure 11 shows an example of the learning result using GNG-DT in Data2. In the topological structure of the position information $C^{pos}$ (Figure 11a), the number of clusters is 1 since the point cloud of Data2 is composed of two objects on the surface plane. In the topological structure of the color information $C^{col}$ (Figure 11b), the number of clusters is 3. The surface point cloud and the part of quadratic prism is clustered as the same cluster since the color is almost the same value. In the topological structure of the normal vector information $C^{nor}$ (Figure 11c), the number of clusters is 4 according to the surface direction. In the topological structure of the color and normal vector information $C^{col} \cdot C^{nor}$ (Figure 11d), the number of clusters is 5 with segmented red and brown planes in the prism by combining the multiple topological structures of the color and normal vector information. In this way, GNG-DT can generate the multiple clustering results according to each property within the framework of the online learning method, preserving the geometric feature of the point cloud.

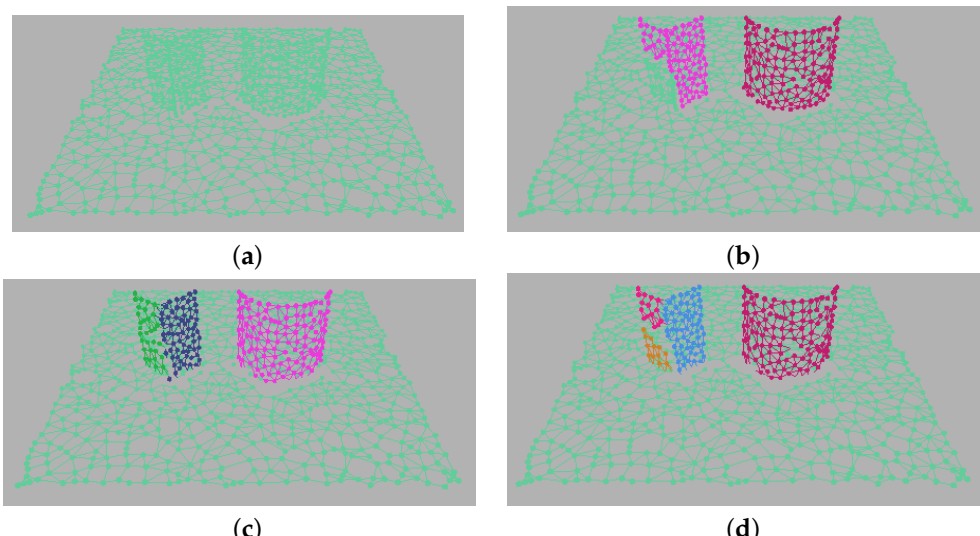

**Figure 11.** Clustering examples in 3D point cloud data (Data2). Each color of the node and edge represents each cluster. (**a**) Position $C^{pos}$, (**b**) color $C^{col}$, (**c**) normal vector $C^{nor}$, (**d**) color $C^{col}$ and normal vector $C^{nor}$.

### 3.3. RGB-D Data

### 3.3.1. Experimental Setup

In this section, we conduct the experiment using 3D point cloud data measured by an RGB-D camera for verifying the effectiveness of the real sensing data. The first column of Figure 12 shows the six datasets used in this experiment. These datasets are measured by Azure Kinect DK and composed of target objects on the floor or desk. In this experiment, we also compare our proposed method with the conventional GNG and GNG-W, and the quantization error is used as the evaluation metric for preserving the 3D position space of the 3D point cloud. The parameters of each method use the same as the above simulation experiment, and the number of trials is 50.

### 3.3.2. Experimental Result

Tables 5 and 6 show the experimental results of the quantization error. In Table 5, GNG-DT outperformed the other methods in all quantization errors of the position information. Next, GNG-W outperformed the other methods in most quantization errors of the color information since the weight of the color information has a positive effect on decreasing the error values. However, the GNG-DT's quantization errors of the color information were smaller than those of GNG-W in (a) and (c). Therefore, the weighted distance based method needs to design suitable weight parameters according to the 3D point cloud measured by an RGB-D camera. On the other hand, GNG-DT preserved the geometric space of the 3D point cloud data accurately, which can be utilized to extract features related to the shape of the point cloud. From the above, the distance using only the position information in GNG-DT is a suitable strategy in the 3D space perception.

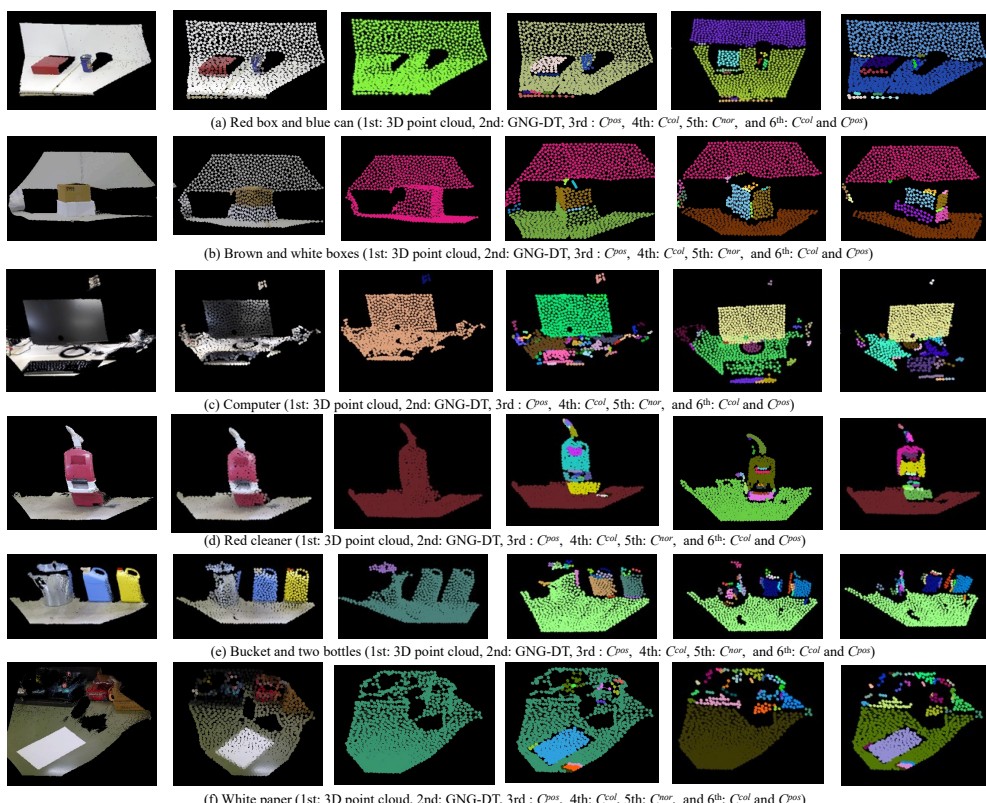

(a) Red box and blue can (1st: 3D point cloud, 2nd: GNG-DT, 3rd : $C^{pos}$, 4th: $C^{col}$, 5th: $C^{nor}$, and 6th: $C^{col}$ and $C^{pos}$)

(b) Brown and white boxes (1st: 3D point cloud, 2nd: GNG-DT, 3rd : $C^{pos}$, 4th: $C^{col}$, 5th: $C^{nor}$, and 6th: $C^{col}$ and $C^{pos}$)

(c) Computer (1st: 3D point cloud, 2nd: GNG-DT, 3rd : $C^{pos}$, 4th: $C^{col}$, 5th: $C^{nor}$, and 6th: $C^{col}$ and $C^{pos}$)

(d) Red cleaner (1st: 3D point cloud, 2nd: GNG-DT, 3rd : $C^{pos}$, 4th: $C^{col}$, 5th: $C^{nor}$, and 6th: $C^{col}$ and $C^{pos}$)

(e) Bucket and two bottles (1st: 3D point cloud, 2nd: GNG-DT, 3rd : $C^{pos}$, 4th: $C^{col}$, 5th: $C^{nor}$, and 6th: $C^{col}$ and $C^{pos}$)

(f) White paper (1st: 3D point cloud, 2nd: GNG-DT, 3rd : $C^{pos}$, 4th: $C^{col}$, 5th: $C^{nor}$, and 6th: $C^{col}$ and $C^{pos}$)

**Figure 12.** Experimental dataset and results of GNG-DT. The first column is the 3D point cloud measured by an RGB-D camera, the second column is the learning results of GNG-DT, the third column is the clustering results using the topological structure of position information $C^{pos}$, the fourth column is the clustering results using the topological structure of color information $C^{col}$, the fifth column is the clustering results using the topological structure of normal vector information $C^{nor}$, and the sixth column is the clustering results using the topological structure of color and normal vector information $C^{col}$ and $C^{nor}$.

**Table 5.** Experimental results of quantization error of position in real environment, where GNG, GNG-W and GNG-DT indicate growing neural gas, GNG with weighted distance measurement, and GNG with different topologies, respectively.

|  | GNG | GNG-W | GNG-DT |
|---|---|---|---|
| Figure 12a | 12.796 ± 0.0011 | 12.341 ± 0.0018 | **11.976 ± 0.0005** |
| Figure 12b | 12.754 ± 0.191 | 12.914 ± 0.015 | **12.754 ± 0.0004** |
| Figure 12c | 11.428 ± 0.001 | 10.857 ± 0.001 | **10.014 ± 0.0005** |
| Figure 12d | 8.908 ± 0.090 | 8.526 ± 0.088 | **8.309 ± 0.109** |
| Figure 12e | 10.151 ± 0.016 | 9.849 ± 0.023 | **9.725 ± 0.043** |
| Figure 12f | 9.492 ± 0.0006 | 9.200 ± 0.0008 | **8.998 ± 0.0005** |

Next, Figure 12 shows examples of the learning and clustering results of GNG-DT. GNG-DT could learn the geometric feature and color information simultaneously. In addition, GNG-DT could provide the different clustering results from the point cloud by using topological structures of each property. In particular, in Figure 12, the topological structure of segment 4 comprises surface planes of white and brown boxes by combining the topological structures of color and normal vector information from the real environmental data. In this way, GNG-DT can be applied to the unknown 3D point cloud measured in the real environment and cluster the point cloud from the viewpoint of multiple properties with online learning.

**Table 6.** Experimental results of quantization error of color in real environment, where GNG, GNG-W and GNG-DT indicate growing neural gas, GNG with weighted distance measurement and GNG with different topologies, respectively.

| | GNG | GNG-W | GNG-DT |
|---|---|---|---|
| Figure 12a | $33.719 \pm 0.715$ | $30.544 \pm 0.650$ | $\mathbf{30.235 \pm 0.221}$ |
| Figure 12b | $14.967 \pm 0.076$ | $\mathbf{13.876 \pm 0.077}$ | $15.392 \pm 0.047$ |
| Figure 12c | $61.171 \pm 0.555$ | $57.318 \pm 0.343$ | $\mathbf{55.773 \pm 0.271}$ |
| Figure 12d | $27.484 \pm 0.322$ | $\mathbf{24.765 \pm 0.232}$ | $26.320 \pm 0.353$ |
| Figure 12e | $18.925 \pm 0.070$ | $\mathbf{17.699 \pm 0.053}$ | $18.642 \pm 0.076$ |
| Figure 12f | $24.535 \pm 0.355$ | $\mathbf{22.426 \pm 0.229}$ | $28.281 \pm 0.268$ |

## 4. Discussion

In the 3D space perception of the autonomous mobile robot, grasping the position between the target object and the robot is an essential capability. Therefore, the accuracy of the position information is the most important for the 3D space perception. In the experiments of the simulation and RGB-D data, GNG-DT performed the other methods, except Data2 (scl = 10.0). This is because the 1st and 2nd winner nodes are selected by using only position information in GNG-DT, which is the important strategy for the 3D space perception. On the other hand, GNG and GNG-W outperformed GNG-DT in the part of the results of the quantization error of the color information. GNG and GNG-W generated some spaces near the boundary of the colors as shown in Figure 8a. On the contrary, GNG-DT generated isolates nodes near the boundary; the color information of these nodes was a mixture of each color information, which affected the quantization error of the color information in the results. However, these isolated nodes represent the boundary of the property, and this information can be utilized for recognizing the shape of the object.

Next, the autonomous mobile robot that performs tasks in the unknown environment needs to recognize the unknown object. For recognizing the unknown objects, the robot needs to extract the invariant of the object by utilizing the multiple properties included in the object. The conventional GNG has only one topological structure, which provides only one clustering result. On the other hand, GNG-DT has topological structures of multiple properties. Therefore, as shown in Figures 11 and 12, GNG-DT can generate multiple clustering results according to the situation and the target object. Furthermore, GNG-DT can generate more clustering results by combining each topological structure. The experimental results of this paper verified the combination of the topological structures by using only the color and normal vector information, but there are many combinations utilizing multiple topological structures. Therefore, we will propose the autonomous cluster generation method according to the situation of the robot by searching suitable combinations from the topological structures.

## 5. Conclusions

This paper proposed growing neural gas with different topologies (GNG-DT) for perceiving the 3D environmental space from unknown 3D point cloud data. GNG-DT can preserve the geometric feature of the 3D point cloud data by using only the position information in the winner node selection and accumulated error calculation. In addition, GNG-DT has multiple topologies of each property for providing the different clustering results according to the properties within the framework of the online learning. The experimental results showed the effectiveness of the GNG-DT by using simulation and real environmental data measured by an RGB-D camera. The performance of the GNG-DT is much better than that of the other conventional methods. In addition, the clustering results showed that GNG-DT can provide various clustering results by utilizing each topological structure.

In this paper, GNG-DT applied to only static data measured by an RGB-D camera. However, we should verify the dynamic data for realizing 3D space perception since the

autonomous robot needs to perceive the 3D environmental space in real time. Therefore, we will propose the learning algorithm for applying dynamic data.

**Author Contributions:** Conceptualization, Y.T.; methodology, Y.T. and A.W.; software, Y.T. and A.W.; validation, Y.T., A.W. and H.M.; formal analysis, Y.T.; investigation, Y.T.; resources, H.M.; data curation, K.O.; writing—original draft preparation, Y.T.; writing—review and editing, Y.T.; visualization, Y.T.; supervision, T.M. and M.M.; project administration, Y.T.; funding acquisition, Y.T. All authors have read and agreed to the published version of the manuscript.

**Funding:** JSPS KAKENHI Grant Number 20K19894.

**Institutional Review Board Statement:** Not applicable.

**Informed Consent Statement:** Not applicable.

**Data Availability Statement:** Not applicable.

**Acknowledgments:** This work was supported by JSPS KAKENHI Grant Number 20K19894.

**Conflicts of Interest:** The authors declare no conflict of interest.

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
