# Peer review of "Growing Neural Gas with Different Topologies for 3D Space Perception"

_applsci, doi:10.3390/app12031705_

Round 1

Reviewer 1 Report

This manuscript provides the Growing Neural Gas with Different Topologies (GNG-DT) for perceiving the 3D environmental space from the unknown 3D point cloud data.
This manuscript has a good overall composition and is well written. However, there is a lack of comparative analysis of the performance of various related techniques and proposed techniques mentioned in the introduction. Additional analysis is needed in the experimental part of the manuscript.

Author Response

Thank you very much for your time and effort to review the paper. We would like to reply to your valuable comment.

Reviewer 2 Report

This paper proposed the Growing Neural Gas with Different Topologies (GNG-DT) for perceiving the 3D environmental space from the unknown 3D point cloud data. GNG-DT applied to only the static data measured by the RGB-D camera.

  • The results and conclusion are missing in the abstract. The introduction section and background should be summarized.
  • For an easier read of the manuscript, the authors should respect different sections of the journal guideline  (Introduction, Material and Methods, Conclusion, Discussion, and Conclusion).
  • Differents studies are conducted about 3D space perception, what is the novelty of your study?
  • Weak literature research ( only 35 references)
  • The whole of references (35) is used only for the Introduction and Background sections.
  • There is no Discussion section? The results were not discussed in the manuscript.
  • What is the implication of this study?

Specific comment:

  • For all tables: PLZ add the abbreviation and their meaning below each corresponding table. 
  • Figure 2 should be clarified.
  • Some equation explications were missed.

Author Response

(The authors gave the same response as above.)

Round 2

Reviewer 1 Report

The revised manuscript is written by reflecting the comments of the reviewers well.

Reviewer 2 Report

The authors took into account all the suggested corrections. Thus, their manuscript is much more comprehensive and accurate.  I consider that the paper is worthwhile to be published in this form.